# “Making Data the Drug”: A Pragmatic Pilot Feasibility Randomized Crossover Trial of Data Visualization as an Intervention for Pediatric Chronic Pain

**DOI:** 10.3390/children10081355

**Published:** 2023-08-07

**Authors:** Katelynn E. Boerner, Unma Desai, Jessica Luu, Karon E. MacLean, Tamara Munzner, Haley Foladare, Jane Shen, Javed Gill, Tim F. Oberlander

**Affiliations:** 1Department of Pediatrics, BC Children’s Hospital Research Institute, University of British Columbia, Vancouver, BC V6H 3V4, Canada; katelynn.boerner@bcchr.ca (K.E.B.);; 2Department of Computer Science, University of British Columbia, Vancouver, BC V6T 1Z4, Canada; 3BC Children’s Hospital, Vancouver, BC V6H 3N1, Canada; 4School of Population and Public Health, University of British Columbia, Vancouver, BC V6T 1Z3, Canada

**Keywords:** chronic pain, pediatric, data visualization, feasibility

## Abstract

Data tracking is a common feature of pain e-health applications, however, viewing visualizations of this data has not been investigated for its potential as an intervention itself. We conducted a pilot feasibility parallel randomized cross-over trial, 1:1 allocation ratio. Participants were youth age 12–18 years recruited from a tertiary-level pediatric chronic pain clinic in Western Canada. Participants completed two weeks of Ecological Momentary Assessment (EMA) data collection, one of which also included access to a data visualization platform to view their results. Order of weeks was randomized, participants were not masked to group assignment. Objectives were to establish feasibility related to recruitment, retention, and participant experience. Of 146 youth approached, 48 were eligible and consented to participation, two actively withdrew prior to the EMA. Most participants reported satisfaction with the process and provided feedback on additional variables of interest. Technical issues with the data collection platform impacted participant experience and data analysis, and only 48% viewed the visualizations. Four youth reported adverse events not related to visualizations. Data visualization offers a promising clinical tool, and patient experience feedback is critical to modifying the platform and addressing technical issues to prepare for deployment in a larger trial.

## 1. Introduction

Pediatric chronic pain is a highly prevalent and distressing condition that affects 3–5% of young people and is associated with significant costs to society and the healthcare system [1,2]. Undertreated pediatric chronic pain is associated with negative social, emotional, developmental, academic, and recreational functioning in childhood and adolescence, and is highly predictive of continued disability and pain in adulthood [3,4,5,6]. Chronic pain is also challenging to assess and treat, as it is based on the individual’s subjective perception of their symptoms and is a highly variable experience, both between- but also within-person [7].

The prevalence and cost of pediatric chronic pain is staggering when compared to the significant lack of available resources to treat it. Providers and centers specialized in providing this care are difficult to access at the best of times, resulting in substantial inequities in who can receive care [8]. Primary care providers have advocated for digital systems to track and monitor their patients’ pain, though with concerns about how to equitably deliver this to underserved communities [9]. This need intensified during the COVID-19 pandemic, where youth with chronic pain described increased challenges in managing pain and mental health, as well as interruptions in accessing pain services [5]. The pandemic highlighted the need for further attention to digital health tools as a potential option for increasing access to pain care in young people, particularly those for whom existing care solutions are inaccessible. Many such tools are available, however, there is a lack of digital health tools that address patient-identified priority areas (e.g., sleep), and existing pain tracking apps are of low quality [10]. Many digital health interventions are not developed with youth or consider developmental needs [11], and even well-designed applications based on evidence-based interventions suffer from difficulties with implementation and sustaining engagement [12].

Tracking symptoms is a major feature within many self-management pain applications. Symptom tracking using an Ecological Momentary Assessment (EMA) approach offers a method of intensive longitudinal data collection that involves repeated sampling of participant experiences in their real-world environment [13]. Such an approach is often referred to as “micro-longitudinal”, as it involves data collection over time, but on a smaller scale than most traditional longitudinal studies (i.e., over days or weeks, rather than months or years). EMA has the benefit of offering more accessible opportunities for real-world data collection, ecologically valid assessment, and avoiding recall bias [14,15]. EMA is a valid and reliable method of collecting data related to chronic pain [16], and has been used to capture temporal relationships between variables [17], evaluate pain trajectories and interference in cancer [18], validate theoretical models of chronic pain [19], and examine mechanisms of change over time [18,20]. 

EMA provides rich data for research teams, but this data is rarely fed back to patients and families despite their significant investment in data collection. Recent patient engagement work by our team highlighted that having a reciprocal relationship with a research team was highly valued by patients and families, with many specifically requesting the ability to access and interact with their own data collected as part of research [15]. A recent trial in adults with low back pain found that participants were unable to reliably recognize the pattern of their pain experience through regular tracking alone, illustrating the potential benefit of being able to engage with visualizations of their pain data [21]. The authors also noted substantial variability within pain trajectories [21], suggesting that the addition of other variables (e.g., sleep, emotions) may support understanding what contributes to this variability in lived experience. Viewing patterns of daily experience data could be inherently therapeutic, sparking new insights, motivating behaviour change, and supporting communication. This could parallel or support evidence-based treatments such as Cognitive-Behavioural Therapy, where patients use patterns of thoughts, feelings, and behaviours to interpret lived experience, encourage functioning, and reframe pain-related cognitions [22]. As such, we hypothesize that interacting with one’s own visualized data could serve as an intervention, “making data the drug” [23].

While EMA has been widely used in children and adolescents [24,25] and pediatric chronic pain populations specifically [17,26], no research to date has examined the impact of visualizing and viewing EMA data in real-time. The aim of this pilot feasibility trial was to determine whether collecting EMA measures of pain and related experiences and engaging with data visualizations would be usable, accessible, and feasible in youth with chronic pain. The present study represents Phase II of the ORBIT model for behavioral treatment development, necessary to establish proof-of-concept and feasibility prior to embarking on an efficacy trial [27]. The visualizations were developed over the course of a substantive, collaborative, 1-year, patient-engaged design process, led by the human-computer-interaction and visualization expert team members with close input from the clinical expert members. The design process, the visualizations themselves and their deployment with a health-tech partner on a secure smartphone platform are described in detail in Desai et al. [28], including description of technical challenges that were encountered that may have impacted participant experience and intervention fidelity. 

## 2. Materials and Methods

This pilot feasibility study was a parallel randomized, single-center, open-label crossover trial with a 1:1 allocation ratio and an exploratory framework. A CONSORT checklist is available as Appendix A. This research was approved by the University of British Columbia Children’s and Women’s Research Ethics Board (H20-02965). 

### 2.1. Setting and Participants

#### 2.1.1. Setting

Recruitment and data collection was conducted out of the Complex Pain Service at a large tertiary-care pediatric hospital in Western Canada. The pain service treats chronic, complex, and persistent non-cancer pain in young people up to age 18 from across the province. To accommodate the vast geographic area covered by the clinic, all aspects of the study from initial invitation through data collection occurred remotely, from the participants’ homes. 

#### 2.1.2. Eligibility Criteria

Youth between the ages of 12–18 years were eligible to participate if they had any type of pain that had persisted for >3 months. Youth and parents were required to have sufficient command of English to participate in consent/assent processes, and for the youth to engage in study tasks. Youth with cerebral palsy, autism spectrum disorder, learning disabilities, attention-deficit hyperactivity disorder, or a genetic/metabolic disorder that interferes with their ability to complete the tasks required for the study were not eligible to take part. Our team is separately exploring accessibility considerations related to engaging these populations in in-home and visualized data research [15]. 

#### 2.1.3. Recruitment and Consent

Participants were recruited from a list of current and former patients of the Complex Pain Service. Participants were first sent a letter of invitation, with the option to opt-out of further contact, and then follow-up occurred by phone or email by a researcher not involved in their clinical care. Families who were interested in participating reviewed the study with a member of the research team and were sent consent (parent) and assent (youth) forms to complete using REDCap [29]; both were required to complete their forms prior to commencing the study. Data collection took place from June to August 2022.

#### 2.1.4. Power Calculation

A target sample size of 50 participants was selected to estimate a retention rate of 80% (95% CI = 69–91%), and as per published recommendations [12]. 

### 2.2. Ecological Momentary Assessment, Visualizations, and Smartphone Deployment

The EMA protocol was developed based on the existing literature on momentary assessment in pediatrics [24,25], and adapting questions from an existing pediatric anxiety EMA protocol [30]. Attention was paid to including questions important in self-management and psychological therapies for pediatric chronic pain, including self-report of emotions, somatic symptoms, interactions with peers, context, and sleep. 

Data was visualized using standard charts such as line charts, bubble charts, bar graphs, heat maps, and custom-designed visualizations to represent multiple aspects of data in one image (see sample in Figure 1). The process of development and user evaluation for the data visualization applications is described elsewhere [28]. Briefly, the visualizations were developed following expert interviews, task mapping, low-, medium-, and high-fidelity prototyping, a usability pilot study, prototype modification, a preliminary utility study, design assessment, implementation into a data capture platform provided by our industry partner CareTeam [31], and piloting prior to deployment in the present study. The prototype was developed through a process of iterative refinement involving experts in pediatric pain, computer science and human-centered design, and lived experience. The aim was to present data in a way that would be understandable and engaging to youth, allowing for examination of trends in their experiences across multiple domains [28].

For the present study, both the EMA and visualization were delivered using the CareTeam platform [31]. Participants received prompts on their smartphone to log in to the CareTeam web-based platform where they could complete questionnaires and (during Part B) view their visualizations. Prompts were delivered 3 times a day, using fixed time-based sampling and a coverage model. 

### 2.3. Procedures

This protocol was pre-registered on Open Science Framework at osf.io/hqx7c on 25 October 2021, and published as a manuscript [23].

Following consent and assent, youth completed baseline questionnaires. They were then randomized as per the protocol detailed below. The trial used an A-B crossover design whereby participants were randomly assigned to first receive either one week of Part A (EMA alone) or one week of Part B (EMA + visualization), followed by a 1-week washout period, and then completed the opposite phase. After each phase, youth evaluated their experiences via a questionnaire, with a subset taking part in a post-trial interview probing more in-depth about their perceptions of the visualizations [28]. We opted for the randomized crossover design to determine whether this approach was feasible to potentially be applied to a larger trial, given the lengthy and intensive process of data collection required. The comparator of EMA alone was chosen to be able to isolate the specific effect (in a larger study) of tracking and entering data on daily pain experiences. In a way, this provides an attention control where the participant continues to be actively engaged in potentially therapeutic self-monitoring through data collection.

Smartphones were offered to youth who needed them for the duration of the study, though all participants either had access to a smartphone or completed the study using a web browser on another device (e.g., tablet or computer). 

#### 2.3.1. Randomization

Randomization was performed using a computerized random number generator. To approximate a 1:1 allocation ratio, participants who received an odd number were assigned to Part A followed by Part B, and even numbers to complete Part B followed by Part A. No blocking or stratification was used. Allocation was concealed as the research assistant conducting the randomization did not use the random number generator for any participant until they had consented and were ready to commence the trial. 

Participants were not masked (please note that the term “masked/masking” is used throughout as an alternative to the traditional term “blinded/blinding” for the purposes of using inclusive language.) to condition, as they had to be aware of whether they were on a dashboard week to know to check in on their visualization dashboard. At the start of each data collection week, participants were sent an email the day before with instructions on how to sign up, how to complete the EMA, and, for youth who were completing Part B, instructions on how to access the dashboard and interpret the visualizations. 

#### 2.3.2. Part A: EMA

Participants completed the EMA protocol 3 times a day for 7 consecutive days on the CareTeam platform. Prompts were delivered in the morning (8:00 a.m.), afternoon (noon), and evening (6:00 p.m.).

#### 2.3.3. Part B: EMA + Data Visualization

Participants completed the same EMA protocol as in Part A, but also had the option of viewing the data visualizations, which were updated in real-time as new data was added. 

#### 2.3.4. Technical Issues during Deployment

After the first week of data collection finished, it came to our attention that some participants were still having difficulty locating the visualization dashboard, therefore, we instituted a reminder email mid-week regarding how to check the dashboard, and CareTeam also implemented an additional pop-up check-in on the platform for participants to indicate whether they had looked at their dashboard. Throughout the study, we continued to receive reports from participants of issues with the CareTeam platform. This was despite researcher pilot testing and a lengthy development process in close collaboration with the platform developers. Minor issues were resolved quickly, and those issues that could not be resolved during the study were flagged for future developments [28]. The most concerning to the team were reports from participants that it appeared that some of their data entry responses were not being saved by the platform. All reports of issues were logged and investigated by the research team and CareTeam.

These technical issues compromised some aspects of the results we aimed to collect: namely, that we could not be certain that all data entered by participants was correctly piped into the visualizations they viewed. Participant reports suggested that some data was not being saved, and a post-hoc review of the dataset revealed some data that was matched to the wrong participant number. Both issues would have potentially impacted the appearance of the visualizations. Such issues related to deployment are a major challenge in digital health studies, and despite these challenges there was still important feedback collected related to feasibility, acceptability, and technical aspects related to deployment. Please see Desai et al. for a more in-depth discussion of these issues [28]. 

#### 2.3.5. Deviations from Pre-Registered Plan

For feasibility of remote administration, instead of using sealed envelopes, randomization was conducted for each participant as they were enrolled, using a computerized random number generator. This essentially replicated the desired 1:1 allocation ratio but was more feasible for administration during the COVID-19 pandemic, where participants and staff were interacting remotely.

Related to the issues of data collection as described herein, the secondary outcomes examining relationships between EMA-measured variables to determine preliminary efficacy were not conducted. Additionally, we opted to not examine completion rates in relation to demographic or pain-related factors, as this would likely have been confounded by the technical issues encountered. See also Section 2.5.1. regarding deviations to the plan to conduct data analysis masked to condition. 

### 2.4. Measures

#### 2.4.1. Baseline Assessment

At baseline, youth provided demographic information regarding age, sex, gender, ethnicity, language, socioeconomic status, pain location, and current treatments. Additionally, the following information was collected via self-report questionnaires administered on REDCap:Pain duration, intensity, and interference were measured with the PROMIS (Patient-Reported Outcomes Measurement Information System^®^) self-report youth pain scales, version 2.0 8a [32,33].Anxiety and depression were measured using the PROMIS Emotional Distress scales, pediatric self-report version 2.0 8a [34].Somatic symptoms were evaluated using the Children’s Somatic Symptom Inventory-8 item version [35].

#### 2.4.2. Feasibility Outcomes

The following feasibility outcomes were measured:Recruitment rate: Number who agreed to participate of the total number eligible, including reasons for declining participation.Retention rate: Number of participants who completed the entire trial of the total who consented.Data completion rate and timeliness/duration of completion: Number of EMA data points completed by participants during the trial, whether EMA ratings were completed during the timeframe (versus back-filling), and length of time spent completing the EMA.Participant ratings of acceptability and feasibility: Investigator-created questionnaire administered at the end of each week.Participant reports of barriers and adverse events: Log of issues reported to the study team and distress reports from the post-week questionnaire.Engagement with visualization dashboard: Number of participants who reported viewing the visualizations of the total who participated.Participant ratings of data visualization: Questionnaire completed by the subset of participants who took part in post-trial interviews; results reported in [28].

No changes were made to the measurement methods after the pilot trial commenced. However, due to limits with the data available from our industry partners we were unable to obtain objective measurements of length of time spent completing the EMA and the number of times participants accessed the dashboard. These may be important variables to assess for future research, though we did assess participant perceptions through proxy variables (i.e., acceptability of the number of EMA questions, self-report of data visualization viewing). 

### 2.5. Data Analysis

#### 2.5.1. Masking to Condition

While the pre-registered protocol described a plan to mask the data analyst to participant condition, it was determined that masking was no longer necessary as we opted not to examine preliminary efficacy outcomes. Instead, issues related to database structure and management were logged to be addressed for a future larger trial, to facilitate masked analysis and reduce the need for post-hoc database restructuring. 

#### 2.5.2. Interim Analysis and Stopping Guidelines

The study was stopped when we had reached close to our target sample size (*n* = 48 of a desired *n* = 50) and when it became clear that significant modifications were needed to the platform; see Section 2.3.4.

#### 2.5.3. Feasibility Outcomes

Descriptive statistics (e.g., means and percentages) were used to calculate most quantitative variables. Content analysis [36] was used to examine common themes reported by participants on the open-ended questions. 

## 3. Results

### 3.1. Demographics and Baseline Characteristics

The following demographic data is reported for all individuals that enrolled and did not actively withdraw prior to starting the trial (*n* = 46). The sample was reflective of the typical demographics of the Complex Pain Service clinic, with the majority of participants identifying as female, English-speaking, European or Asian origins, and comfortable socioeconomic status (see Table 1). Participants were engaged in variety of current treatments for their pain condition, with the most common being pharmacological treatments, physiotherapy, and psychology/counselling. On the baseline questionnaires (see Table 2), participants reported musculoskeletal pain as the most common type of pain, which had lasted an average of 4 and a half years, with an average intensity of 5.5 out of 10. Average pain interference was >1 SD above the mean. 

### 3.2. Data Cleaning

The original data file received from CareTeam contained 1410 responses (where a response indicates that the participant completed at least 1 data field on a particular EMA survey), including test records and duplicates. Among those, 179 responses were impacted by an error in matching the data to the correct participant number. 105 were able to be successfully rematched, representing 36 separate participants (a range of 1 to 7 responses filed incorrectly for these individuals). If the error occurred while the participant was completing Part B (EMA + data visualization), as was the case for 47 responses representing 25 participants, the data may have either shown up as blank/missing, or incorrect data may have appeared on the visualization dashboard, as the rematching did not occur until after data collection was finished and the error was identified. Of the 25 participants affected, only 12 indicated that they had accessed the dashboard during Part B. 

Following the rematching, 33 responses were removed as they represented test records, and 17 duplicate entries (i.e., according to the survey due date the survey was filled in twice) were removed, with the first logged entry being kept. 

### 3.3. Primary Feasibility Outcomes

#### 3.3.1. Recruitment Rate 

Of those assessed for eligibility, 37% of individuals agreed to participate in the study (see Figure 2). The vast majority of those who declined were either did not respond to study invitations, or indicated interest but then were lost to follow-up. Of those who provided a reason for declining, four families specifically expressed that they did not want to be reminded of pain three times a day for two weeks by taking part. One family reported smartphone access as one (of several) reasons for declining to participate. 

#### 3.3.2. Retention Rate 

Of the 48 enrolled participants, 2 withdrew prior to beginning the EMA, resulting in a 96% retention rate. An additional 2 participants did not actively withdraw, but did not sign up for the CareTeam platform or complete any study activities beyond the baseline questionnaire. Other families had varying levels of engagement, including some who participated in portions of the EMA but did not provide feedback on their experience (see Figure 2 and Section 3.3.3 for more details). Families cited reasons such as other commitments, travel, and unexpected medical events as reasons for lack of participation.

#### 3.3.3. Data Completion Rate 

We received at least one response (i.e., at least one EMA entry with at least 1 data field filled in) from 43 participants, representing 93% of participants who had not actively withdrawn. There were 1360 completed responses for the entire trial, representing 70.4% of 1932 possible responses (46 enrolled participants × 14 days × 3 EMA/day). Completion rates were consistent across the morning (73%), afternoon (70%), and evening (69%) data collection points, and across weeks where participants could access the visualization (69%) compared to EMA alone (72%).

The majority of the EMA prompts were completed within 1 day of the prompt being sent. For the purposes of the present study, back-filing was calculated as being ≥2 days after the due date (to account for issues related to how the system assigned the due date). Backfilling was estimated to occur for 3.8% (18/468) of the morning responses, 4.0% (18/448) of the afternoon responses, and 2.3% (10/444) of the evening responses. 

#### 3.3.4. Participant Ratings of Acceptability and Feasibility 

Ratings of the non-visualization aspects of the trial were generally similar across Part A and Part B, and are reported in Table 3. Generally, participants reported being satisfied with many aspects of the study, including the number of prompts and questions and associated compensation. 

#### 3.3.5. Participant Reports of Barriers and Adverse Events

Many participants reported issues related to technical aspects of the platform, primarily that their EMA responses appeared not to save after having been completed (see Table 3). Other issues were reported intermittently (e.g., having forgot or been too busy to complete all time points, issues related to access to data or WiFi to log in to the platform).

Four participants indicated the presence of emotional distress on their follow-up questionnaires and, as per protocol, a member of the research team followed-up with the participant in each instance. No further action was needed, and all participants agreed to continue with the study. Three distress reports occurred following Part A (EMA only), and two following Part B (EMA + visualization). There was also an equal distribution of distress reports at the end of the first and second week of data collection. Reasons for distress included EMA questions serving as a reminder of past challenging experiences (*n* = 1), study participation contributing to stress/overwhelm (*n* = 2), and reasons unrelated to study participation (*n* = 1). 

#### 3.3.6. Engagement with Data Dashboard 

Of the participants who completed their visualization-related follow-up questionnaires at the end of the Part B week (EMA + visualizations), only 48% reported having viewed the visualizations. See visualization feedback in Table 4. 

#### 3.3.7. Participant Ratings of Data Visualization Use 

Of the participants who did look at the visualizations, the majority described finding it interesting or helpful to look at trends in their data. Others reported finding it useful to have access to the data to support their memory of experiences, and some referenced specific variables that they found helpful within the data (e.g., the comparison of expectations/worries compared to objective outcomes, sleep tracking). Suggestions were made to improve or clarify the visual presentation of the data, and several participants referenced issues related to the technical challenges of data not saving properly. One participant reported that they did not find looking at the visualizations useful to them.

Of those who did not look at the visualizations when they were available, the majority reported that the primary barrier was related to difficulty navigating the platform to find the visualizations. Approximately a third indicated that they forgot or did not prioritize looking at the visualizations, and one participant indicated that they were not interested.

## 4. Discussion

This pilot study reported the feasibility of using EMA data collection to enable youth with chronic pain to track and visualize their experiences in real-time. Ratings of feasibility and acceptability were generally consistent across the two conditions, as were EMA completion rates, suggesting that having access to the visualizations did not improve engagement with the EMA protocol. These results should be interpreted with caution given that a substantial proportion of youth reported not having accessed the visualizations during their Part B week. Participants reported appreciating the opportunity to reflect on and express their experiences through the EMA protocol, and finding that the repeated measurement presented opportunities to observe patterns in their experience. However, many reported burden related to the time it took, technical issues, and repetition. Most participants found the visualizations useful for supporting their understanding of trends in their experience, however, many had feedback regarding technical issues encountered and some participants were unable to locate the visualizations on the platform. Adverse events were rare. Measuring adverse events was a strength of the present study, as these variables are not typically reported in research on pediatric chronic pain psychology interventions or digital health tools [38,39].

The present sample was representative of the demographics and baseline outcomes of youth at other Canadian tertiary-care pediatric pain clinics [15]. Recruitment and retention data suggests that a larger trial could be feasibly accomplished within a relatively prompt timeframe. 

### 4.1. Limitations

As access to the data visualization component was self-selected, and usage of the visualization was collected only by self-report, we cannot confirm that all youth in this study received the same “dose” of the visualization. In fact, engagement rates would suggest that this was not at all the case. In this trial we encountered significant issues with our technical platform that resulted in many youth having difficulty consistently accessing their visualizations, and potential data loss that may have impacted the visualizations that were seen if they were viewed [28]. Future development of the platform would benefit from resolving these technical issues, involving a system of tracking youth engagement with the visualizations, and having a method to capture what youth were viewing at any given time to cross-validate if needed. 

The technical issues also impacted youth’s experience of completing the EMA, which was raised by many in their feedback questionnaires. Despite this frustration, youth continued to persist with data collection with acceptable compliance rates, speaking to their level of engagement in the study. 

Finally, as the present study was not powered to detect efficacy, this will need to be the focus of a larger randomized controlled trial to determine whether these ‘pictures of pain’ can be used to improve pain symptoms and daily function in youth with chronic pain.

### 4.2. Lessons Learned and Recommendations for Future Trials 

While we believe that our data suggests the feasibility, acceptability, and potential of using a combination of EMA and data visualization to research and support youth with chronic pain, several technical issues with our platform and design limit the extent to which conclusions can be drawn from this pilot trial. There are recommendations that we put forward for future data visualization studies, and that would be necessary to rectify before progression to a larger efficacy trial:Trial timing may have impacted the findings as most recruitment and data collection occurred during the final exam period and summer holidays for adolescents in Canada. While this may have offered more flexibility in participant schedule, it also made some aspects of recruitment and maintaining engagement more challenging when participants were in high-stress periods, had more unstructured time, or were travelling for the holidays.A thorough orientation to the platform is likely to be necessary, given that many youths reported being unable to access the visualizations. Additionally, improved clarity around visualization access and/or built-in prompts to access visualizations may support youth navigation within the platform.Participant feedback suggested that are other important variables that participants would wish to track. While many of the EMA domains largely mapped on to established core outcome sets for pediatric chronic pain clinical trials [40], there is an opportunity to further tailor this based on variables of interest for youth with chronic pain. Information about diet, other symptoms, menstrual cycle, and other pain triggers were reported as being variables of interest for youth to track.Even with an established platform, more thorough testing of the data pipeline before implementation, including checking of the export and collection of data processes, may have mitigated some of the matching and data loss issues encountered later.

For a more in-depth examination of the technical development and deployment of the platform, as well as mitigations of the challenges encountered, please see Desai et al. [28].

### 4.3. Clinical Implications

A data visualization platform offers many potential benefits for enhancing the effects of existing pain therapies, such as tracking and motivating adherence to a physical activity program, or providing supporting data for cognitive-behavioral interventions (e.g., graphs depicting fears/expectations compared to reality). This is particularly relevant as research on digital mental health tools demonstrated that effects were only seen when participants had access to regular interactions with a therapist, compared to solely self-directed interventions [41]. Further research is needed to examine opportunities for integration of this tool within multidisciplinary pain care.

### 4.4. Future Directions

This feasibility study is a first step to determine the potential of data visualization of repeated self-reports of pain, anxiety, somatic symptoms and social experience as a novel treatment for youth with chronic pain. The concept of data visualization as an intervention for pediatric pain is a promising one, though its success will be highly dependent on the technology deployment, particularly how engaging and user-friendly the interface is [41]. Future research may consider how the visualization of the data could be enhanced using artificial intelligence programs to provide more sophisticated representations of pain experiences (using, for example, systems such as DALL-E2, which creates images based on natural language descriptions [42]). Integration of EMA data collection with another modality may offer the opportunity for more data with less effort on the part of the participant, such as activity monitoring to support behavioural activation and pacing, as has been explored in adults [43].

It is debatable whether tracking pain is itself counter to typical pain management interventions where attempts are made to redirect attention away from pain and towards functioning. Indeed, while adverse events were rare in the present study, some participants reported distress related to the monitoring of pain experiences. Individual experiences of perfectionism or trauma, both common in pediatric chronic pain [44,45], may impact the extent to which participants engage with and benefit from EMA data collection and visualization. Future research would be required to examine whether participant expectations of benefit and experience influence engagement, data visualization, and outcomes [46,47], and whether the framing of the EMA questions and subsequent visualizations could be used to promote a growth or self-management mindset [48,49].

Emerging research has used EMA to determine individual patient profiles that may inform the use of specific therapeutic techniques [17,50]. Future iterations of this platform may incorporate this approach to delivering personalized intervention suggestions, the effect of which could then be tracked using the visualization component. Research is needed on how to customize EMA delivery coupled with visualizations based on individual user characteristics [51,52], potentially incorporating machine learning and dynamic classification testing, as well as patient and family engagement. More research is also needed to understand how patients use and infer patterns from their data, and whether there are ways to support understanding of meaningful relationships vs. spurious correlations.

Accessibility is a significant limitation of the existing digital health literature on chronic pain [8,10]. Our team is currently developing a version of this data visualization platform specifically for autistic youth, in response to patient-identified priorities [15]. Relatedly, as many evidence-based pain management applications are not publicly available beyond the funding period of the study [53] future research should take hybrid effectiveness-implementation design approaches to ensure the adoption and sustainability of the approach is being considered [12,54,55], and to consider the integration of such a platform within a larger clinical digital health strategy or data pipeline [56]. 

## 5. Conclusions

The present project suggests the feasibility of a larger trial examining data visualization as an intervention for pediatric chronic pain. Such trials have the potential to address a number of patient-oriented research priorities in pediatric chronic pain [57], both with respect to research (e.g., providing a platform to examine the efficacy and timing of treatments, and the interactions between chronic pain and mental health) as well as improving health care delivery (i.e., providing a strategy that may improve access to care and reduce disparities in access to care). 

## Figures and Tables

**Figure 1 children-10-01355-f001:**
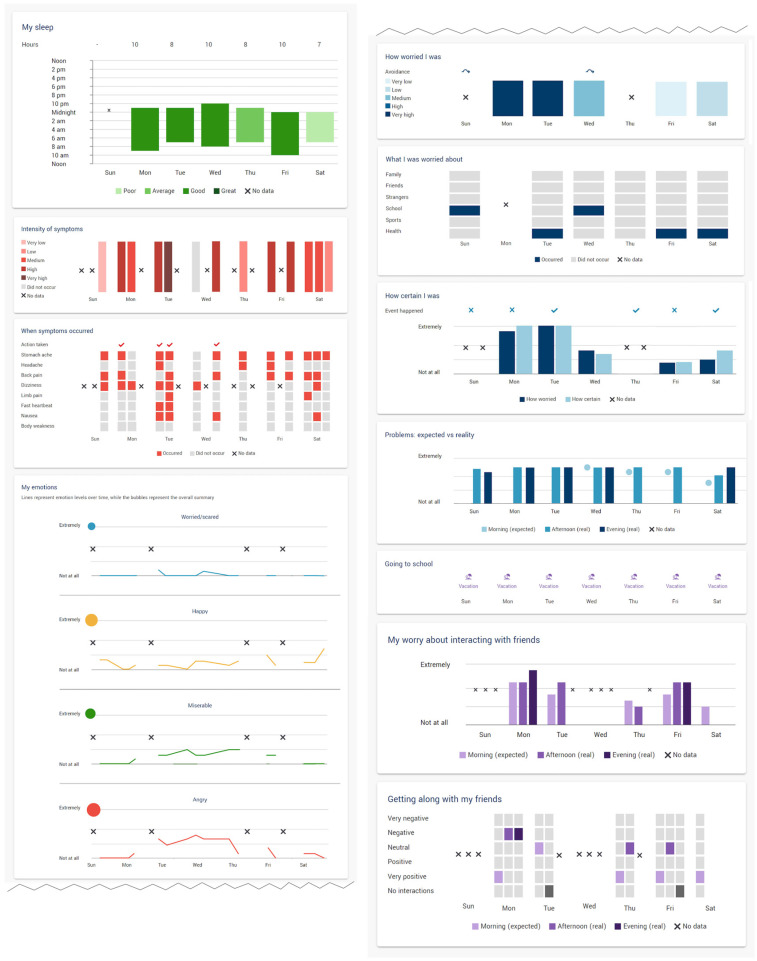
Depiction of the visualization interface as deployed in CareTeam, with composite examples of participant data populating the visualizations. Similar figures and more in-depth presentation available in Desai et al. [28].

**Figure 2 children-10-01355-f002:**
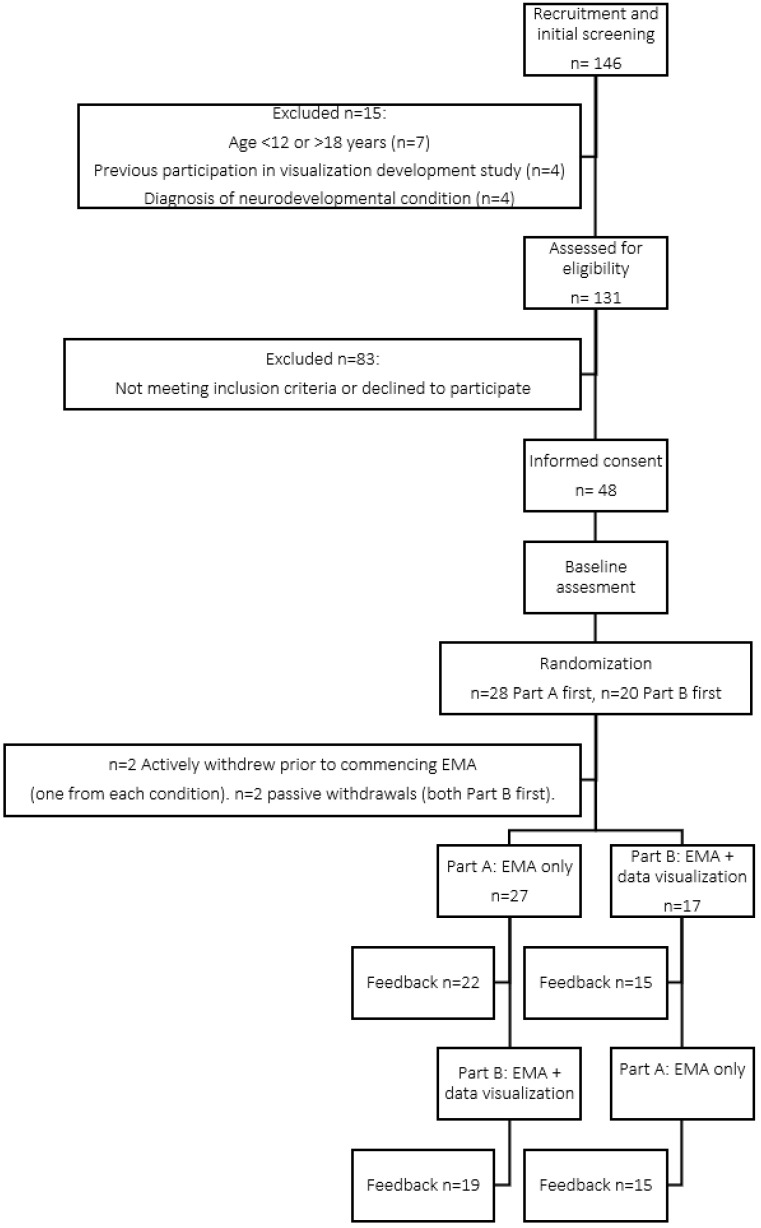
Flow chart of study participation. A subset of participants were invited to complete an additional in-depth interview and questionnaire regarding the visualizations at the end of the study, results described in Desai et al. [28].

**Table 1 children-10-01355-t001:** Demographic characteristics for analyzed sample (*n* = 46).

	N (%) or M (SD)
Age (years)	15.63 (1.73) *Range* = 12–18 years
Sex assigned at birth	
Female	34 (73.9%)
Male	12 (26.1%)
Gender identity ^a,b^	
Female	30 (65.2%)
Male	14 (30.4%)
Indigenous or other cultural gender minority identity	1 (2.2%)
Non-binary	1 (2.2%)
Born in Canada	43 (93.5%)
Languages spoken at home ^a,b^	
English	45 (97.8%)
Other ^c^	8 (17.4%)
Ethnicity ^a,b,d^	
African origins	1 (2.2%)
Asian origins	8 (17.4%)
European origins	29 (63.0%)
North American Aboriginal origins	3 (6.5%)
Other North American origins	7 (15.2%)
Family socioeconomic status ^b^	
Very well off	6 (13%)
Quite well off	10 (21.7%)
Average	22 (47.8%)
Not so well off	4 (8.7%)
Not at all well off	1 (2.2%)
Current treatments ^a^	
Over-the-counter medication	26 (56.5%)
Prescription medication	18 (39.1%)
Physiotherapy	23 (50%)
Psychology	8 (17.4%)
Occupational therapy	3 (6.5%)
Naturopathic therapy	6 (13%)
Chiropractic therapy	6 (13%)
Massage therapy	21 (45.7%)
Acupuncture	7 (15.2%)
Hot-cold treatments	5 (10.9%)
Herbal remedies	4 (8.7%)
Counselling or other mental health treatments	15 (32.6%)
Other (myoactivation and supplements)	2 (4.3%)

Note. ^a^ Participants could select all that applied. ^b^ At least one participant indicated that they prefer not to respond to this question. ^c^ To protect participant confidentiality, specific languages not reported here. ^d^ Ethnicity options were presented to participants based on the Statistics Canada 2016 Census categories; any participant who responded “Other” had their response categorized into one of the available options based on the Census Dictionary [37].

**Table 2 children-10-01355-t002:** Baseline questionnaire responses for analyzed sample (*n* = 44) ^a^.

	N (%) or M (SD)
Pain	
Location ^b^	
Musculoskeletal	27 (61%)
Back	11 (25%)
Multi-site	8 (18%)
Abdominal	6 (13%)
Headache, migraine	3 (7%)
Other	6 (13%)
Number of pain sites ^c^	2.2 (2.4) Range = 1–12
Duration (years)	4.8 (3.9) Range = 0.5–18
Intensity (past 7 days)	5.5 (2.1) Range = 1–9
Pain interference (PROMIS), *T*-score	61.60 (7.42) Range = 9–38
**Somatic symptoms (CSSI-8)**	**14.59 (6.68)** **Range = 3–29**
**Depressive symptoms (PROMIS), *T*-score ^d^**	**57.53 (9.72)** **Range = 35.2–73.6**
**Anxiety symptoms (PROMIS), *T*-score**	**55.04 (8.74)** **Range = 33.5–69.4**

Note. ^a^ Two participants did not complete this section of the baseline questionnaires. ^b^ Free-text response, coded by investigators; participants could indicate >1 pain site. ^c^ Coded based on free-text response to location question, does not include *n* = 7 participants who reported “everywhere”. ^d^
*n* = 43.

**Table 3 children-10-01355-t003:** Number and percentage of responses to follow-up questions for each condition, including representative quotes for some coded categories of qualitative response.

	Part A (EMA Only) *n* = 37	Part B (EMA + Visualization) *n* = 34
**Quantitative questions**		
*How comfortable did you feel answering the questions on the EMA?*		
Very uncomfortable	8 (21.6%)	4 (11.8%)
Uncomfortable	-	-
Neutral	7 (18.9%)	6 (17.6%)
Comfortable	15 (40.5%)	12 (35.3%)
Very comfortable	7 (18.9%)	12 (35.5%)
*How was it filling out questions 3 times a day?*		
3 times a day was too much	13 (35.1%)	12 (35.3%)
3 times a day was just right	22 (59.5%)	21 (61.8%)
3 times a day was too few times	2 (5.4%)	1 (2.9%)
*How did you find the number of questions asked each time?*		
Too many questions	3 (8.1%)	6 (17.6%)
Just right	30 (81.1%)	25 (73.5%)
Too few questions	4 (10.8%)	3 (8.8%)
*What did you do when you received the reminder if you were in the middle of doing something else?*		
Stopped what I was doing and answered the questions	8 (22.2%)	10 (29.4%)
Finished the tasks I was working on and did the questions after	26 (72.2%)	20 (58.8%)
Did the questions later	5 (13.9%)	5 (14.7%)
Other	6 (16.7%)	8 (23.5%)
*Is the amount that you got paid to participate in this study enough for what we are asking you to do?*		
Yes	31 (86.1%)	32 (94.1%)
**Qualitative (free-text) questions**		
*What did you like about participating in the EMA study this week?* ^a^		
Feasibility, including facilitating routine	11 (29.7%) “I liked how I didn’t have to worry about them because they were fast and easy and i didn’t have much in my day so it followed my schedule”	6 (17.6%) “That it was from home.”
Opportunity for reflection	9 (24.3%) “I am able to think more about how I feel each day. Normally I don’t think of checking up on myself.”	6 (17.6%) “Participating in the survey this week gave me insight to the what triggers my pain and how it affects how I am feeling. The survey also made me reflect throughout the day which helped me feel present during the week. ”
Helping others	4 (10.8%) “I liked participating in this because it helps the study and can help others in the future”	5 (14.7%) “I’m glad that my experiences might help other children and youth in my position.”
Seeing patterns	3 (8.1%) “Participating in the EMA study this week allowed me to reflect on my physical and emotional feelings throughout the day. It helped me notice patterns in my behaviour that show me how I function on a day to day basis.”	5 (14.7%) “I liked how I could document my emotions and have them displayed on a chart. I also liked how I got the option to rate my pain from almost no pain to the worst I experienced. Overall it was eye opening to be able to look back on the data and see my behavior.”
Expressing self	2 (5.4%) “I felt like I had somewhere to vent my feelings and how I felt during the week”	2 (5.9%) “I liked being able to express my feelings”
Tracking	3 (8.1%) “Being able to keep track of my emotions and feelings”	2 (5.9%) “It helped me keep track of which day of the week it was. It was useful to write down my pain. It made me realize how many symptoms I have and experience all the time.”
Others less commonly reported included the visualizations (*n* = 3) and feeling cared for/checked in on (*n* = 2).		
*What did you dislike about participating in the EMA study this week?* ^a^		
Burden	15 (40.5%) “I didn’t like that I had do to i in the car a couple times.”	12 (35.3%) “Although I liked the routine the times fell awkwardly with my work schedule. I felt rushed to complete the noon submission because I was doing it at work”
Technical issues	6 (16.7%) “Sometimes I would do the survey and it wouldn’t save”	11 (32.4%) “However, the system did not register many of the surveys that I actually did fill out and complete on my phone. This also made the graphs difficult to interpret and read because of all the missed data.”
Repetitive nature of questions/procedures	7 (18.9%) “The questions weren’t diverse enough to get different answers out of me some days.”	4 (11.7%) “it was very tedious after a couple days”
Content of EMA ^d^	6 (16.7%) “I disliked that all three surveys were mainly about the thing I was most worried about that day. Often, if the thing I was worried about was a specific event, it had ended before I finished all three surveys. This made it difficult to answer the questions for an event that was already over.”	6 (17.6%) “I sometimes did not feel like the questions applied to me or my life, and I wish there were more about the pain I am experiencing and how it affects me.”
Issues related to WiFi/data access	1 (2.7%) “that you needed wifi to fill out the questions”	2 (5.9%) “The fact you needed to have wifi and if wasn’t like an app this was difficult for people who don’t have data.”
*Did you have any problems in doing the EMA study? If so, please describe them here* (*e.g., technical difficulties, not having access to your smartphone when the text messages arrived, questions too hard to understand, forgot to answer*):		
Technical issues	18 (48.6%)	20 (58.8%)
Too busy/prompts at difficult times to respond	7 (18.9%)	4 (11.7%)
Forgot	5 (13.5%)	4 (11.7%)
Issues related to WiFi/data access, internet outage, etc.	2 (5.4%)	3 (8.8%)
Wording of questions	2 (5.4%)	3 (8.8%)
*What did you think of the timing of the text message reminders?* ^a^		
Good ^c^	23 (62.2%)	19 (55.9%)
Did not receive reminders	4 (10.8%)	3 (8.8%)
Morning prompt (8:00 A.M.) was too early	3 (8.1%)	4 (11.7%)
Afternoon prompt (noon) was too busy	2 (5.4%)	1 (2.9%)
Evening prompt (6:00 P.M.) was too early/late	1 (2.7%)	2 (5.9%)
Did not have access to phone	1 (2.7%)	2 (5.9%)
*Is there anything else we could track with the EMA questions that would help us better understand your pain/your day?* ^a,b^		
Diet	12 (32.4%)	10 (29.4%)
Symptoms (e.g., more details about pain, other symptoms)	8 (21.6%)	11 (32.3%)
Menstrual cycle	7 (18.9%)	11 (32.3%)
Weather	8 (21.6%)	2 (5.9%)
Physical activity	5 (13.5%)	4 (11.7%)
Details about the day (e.g., major events, context like scheduling)	5 (13.5%)	3 (8.8%)
Other less frequent responses included pain triggers, medications, healthcare interactions, other health conditions, stressful events, pain relievers, pain-related cognitions, and opportunities to elaborate on existing topics.		
*What kinds of questions should we be answering with this kind of research?* ^a^		
Relationship between mental health and pain	7 (18.9%)	7 (20.6%)
How to improve care (self-management or provider-led)	7 (18.9%)	6 (17.6%)
Impact of pain on daily life	5 (13.5%)	4 (11.7%)
Characteristics of pain (e.g., frequency, variability, tolerance)	1 (2.7%)	5 (14.7%)
Pain origins and associated triggers	2 (5.4%)	4 (11.7%)
Relationship between sleep/fatigue and pain	2 (5.4%)	2 (5.9%)
Other less frequent responses included understanding the impact of pain on functioning, impact of tracking pain, accuracy in estimating events, and comparing individual responses with population-level		
*Do you have any other questions or feedback you would like to give us about participating in this study?* ^a^		
Enjoyed the experience/found it beneficial	11 (29.7%)	5 (14.7%)
Technical suggestions/issues (e.g., preference for an app than a website, did not receive reminders, no data/WiFi access)	3 (8.1%)	4 (11.7%)
Suggestions or confusion regarding EMA question wording	4 (10.8%)	1 (2.9%)
Did not know how to access mental health resources mentioned by the team	2 (5.4%)	2 (5.9%)
Suggestions regarding timing of prompts	1 (2.7%)	2 (5.9%)

Note. ^a^ Open-ended question coded with qualitative content analysis, therefore more than one concept could have been represented within a participants’ response. ^b^ Coding combines two questions that had significant overlap in responses: “*Was there anything that the EMA questions should have been asking about* (*e.g., different experiences, symptoms, emotions*) *that would have helped us better understand how your day was going?*” and “*Is there anything else you would like to have been tracking with the EMA questions* (*e.g., weather, diet, menstrual cycle, other health conditions*) *that would have helped us to better understand your pain?*” Note that many of the responses incorporated items that were suggested in the question wording. ^c^ In both conditions, many youth indicated that text messages were more helpful than email. ^d^ Content concerns were primarily related to the worries questions (e.g., finding them not relevant) or wanting more questions about specifics of experiences (e.g., pain).

**Table 4 children-10-01355-t004:** Themes of participant responses to open-ended questions about data visualization, with example quotes (*n* = 33) ^a^.

Participants Who Looked at the Dashboard (*n* = 16)	
*Things they liked, found helpful, or useful*	
Looking at trends (56%)	“Being able to visually see how I was doing through the week”
Memory aid (13%)	“Allowed me to see everything at once instead of trying to remember how I felt on a certain day”
Specific variables (13%)	“the expectations vs realty was a good reality check if you were overthinking”
*Things they disliked, found unhelpful or confusing*	
Data display (31%)	“There should be an option to change what type of graph so it is easier to read for some.”
Found it generally confusing (25%)	“It was difficult and confusing to interpret/read with missing data so it looked “blotchy” instead of being a consistent chart.”
Missing data (19%)	“Some of the graph styles were not fitting of the data they were representing, as well as the gaps in the data due to technical error made it really hard to properly read in the case of such a short study”
Not useful (6%)	“I looked at it, but did not really find much use in it. It did not really help me much”
**Participants that did not look at the dashboard (*n* = 17)**	
*Why not?*	
Could not find it/did not know it existed (53%)	“I assume I would love to look at a visual graph of my day, but I couldn’t find my dashboard anywhere so I am still looking.”
Too busy or forgot (29%)	“the amount of time i put in each day felt like enough and it didn’t bother me to not check it regularly”
Did not want to look (6%)	“Because I didn’t want to”

Note. Qualitative coding with content analysis allowed for a single participant to endorse multiple codes, and not all participants answered every question, therefore may not add up to 100%. ^a^ Of the 34 participants who completed Part B questionnaires, one did not answer any questions related to the visualization.

## Data Availability

De-identified quantitative EMA data will be made available at osf.io/hqx7c (accessed on 25 October 2021); availability of de-identified quantitative EMA data was a condition of participant consent and approved by the above named ethics board.

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
