# Peer review of "“Making Data the Drug”: A Pragmatic Pilot Feasibility Randomized Crossover Trial of Data Visualization as an Intervention for Pediatric Chronic Pain"

_children, 2023, doi:10.3390/children10081355_

Round 1

Reviewer 1 Report

Study by Katelynn E. Boerner et al. “Making data the drug: A pragmatic pilot feasibility randomized crossover trial of data visualization as an intervention for pediatric chronic pain”.

 The author provided evidence of the importance of data visualization as a clinical tool, and patient experience feedback is critical in managing and as intervention for pediatric chronic pain.

 Authors used pain e-health application and recruited participants aged 12-18 years from a tertiary-level pediatric chronic pain clinic in Western Canada. Participants completed two weeks of Ecological Momentary Assessment (EMA) data collection and included access to a data visualization platform and enabled participants to view their results. Authors randomized orders of weeks and included 48 participants of 148 youth approached for the research.

 In introduction authors mentioned that 3-5 % of youth suffer from chronic pain(pediatric) and untreated such pain and is associated with negative social, emotional, developmental, academic, and recreational functioning in childhood and adolescence and continued disability and pain in adulthood. Authors mentioned the cost of treating such pain is rising and lack of resources to treat it. Authors proposed the use of digital systems to trac pain and deliver it equitably, especially for those whose existing care solutions are inaccessible. Authors found the gap in virtual care solutions for families of youth with chronic pain illustrated that there is a lack of digital health tools that address patient-identified priority.

Authors explained that tracking symptoms is a major feature within many self-management pain applications, symptom tracking, longitudinal data collection, participant experience in real world environment. Authors mentioned such approaches as micro longitudinal data collection in days and weeks rather than in months and years.

 There is a lack of data feedback to the patients and authors claimed that reciprocal relationship with a research team was highly valued by patients and families and data visualizations has therapeutic values, motivating behavior change, and supporting communication leading to function-promoting actions and reframing pain-related cognitions. Authors therefore hypothesize that interacting with one’s own visualized data could serve as an intervention, “making data the drug”. The aim of this pilot feasibility trial was to determine whether visualization of EMA-collected measures of pain and related experiences would be usable, accessible, and feasible in youth with chronic pain.

 The study was a parallel randomized, single-center, open-label crossover trial with a 1:1 allocation ratio and an exploratory framework. Data collection was conducted out of the Complex Pain Service at a large tertiary-care pediatric hospital in Western Canada. Youth between the ages of 12-18 years were eligible to participate if they had any type of pain that had persisted for >3 months and included those who can complete the tasks required for the study. A target sample size of 50 participants was selected to estimate a retention rate of 80% (95% CI= 69-91%), and as per published recommendations. The trial used an A-B crossover design whereby participants were randomly assigned to receive either one week of Part A (EMA alone) or one week of Part B (EMA + visualization) first, followed by a 1-week and then completed the opposite phase. Participants completed the EMA protocol 3 times a day for 7 consecutive days on the Care Team platform. Prompts were delivered in the morning (8:00am), afternoon (noon), and evening (6pm) to the smartphones used by the participants.

Data collected included pain duration, intensity, interference, Anxiety, depression, and somatic symptoms. Descriptive statistics were used to calculate the most quantitative variables.

 The results collected data for Demographics and baseline characteristic, Recruitment rate (37%), retention rate (96%), data completion rate (93%), Participant ratings of acceptability and feasibility (satisfied), Participant reports of barriers and adverse events (challenging), Engagement with data dashboard -visualization (48%), Participant ratings of data visualization use (helpful and useful).

 This pilot study reported the feasibility of using smartphone approach to EMA data collection to enable youth with chronic pain to record the way they feel and act, then to visualize their symptoms in real-time. Authors concluded that visualizations are useful for supporting their understanding of their experience. Authors do mention the difficulty by many participants with technical issues in accessing their visualizations, and potential data loss that may have impacted the visualizations that were seen if they were viewed. Authors recommended for future visualizations studies such as Trial timing, orientation, and other important variables.

Authors found that data visualization platforms have clinical implication in enhancing effects of existing pain therapies, tracking and motivating adherence to a physical activity program, and cognitive-behavioral interventions.

 The authors concluded that the potential of data visualization of repeated self-reports of pain, anxiety, somatic symptoms and social experience is a novel treatment for youth with chronic pain.

 The study seems fine, methods are written in detail, data are convincing, article language is adequate throughout the manuscript. There are a few corrections needed which are easy to correct.

Line – 114 typo error- delete ‘to’ after through.

Line 181- spell check – ‘lengthy’.

Study by Katelynn E. Boerner et al. “Making data the drug: A pragmatic pilot feasibility randomized crossover trial of data visualization as an intervention for pediatric chronic pain” has clinical significance in treating chronic pain in pediatric population using data visualization and engaging the patients while collecting their own data. Using real time data visualizations as an intervention in managing any chronic pain can be a valuable tool not only for the clinicians but also for the patient with a better outcome. 

Article language is adequate throughout the manuscript. There are a few corrections needed which are easy to correct.

Line – 114 typo error- delete ‘to’ after through.

Line 181- spell check – ‘lengthy’.

Reviewer 2 Report

After completion of the assessment work on this manuscript entitled on '“Making data the drug”: A pragmatic pilot feasibility randomized crossover trial of data visualization as an intervention for pediatric chronic pain', in current form it is unsuitable to recommend for publication due to syntax errors. Hence, minor revision is essential.

Authors have good knowledge regarding the use of Ecological Momentary Assessment (EMA) for data visualization in case of pediatric chronic pain. But minor revision is essential in order to minimize syntax errors as well as scientific explanation of the result of this study.

There are many syntax errors. Language is required to improve using precise concise meaningful sentences in results and discussion, as well as at conclusion portion.
